# A study of deep active learning methods to reduce labelling efforts in biomedical relation extraction

Charlotte Nachtegael[1,2]*, Jacopo De Stefani[2,3], Tom Lenaerts[1,2,4]

**1** Interuniversity Institute of Bioinformatics in Brussels, Université Libre de Bruxelles-Vrije Universiteit Brussel, Bruxelles, Belgium, **2** Machine Learning Group, Université Libre de Bruxelles, Bruxelles, Belgium, **3** Technology, Policy and Management Faculty, Technische Universiteit Delft, Delft, Netherlands, **4** Artificial Intelligence Laboratory, Vrije Universiteit Brussel, Bruxelles, Belgium

* Charlotte.Nachtegael@ulb.be

**Data Availability Statement:** All relevant data and code are within the paper and its Supporting information files, and are available from the GitHub repository (https://github.com/oligogenic/Deep_active_learning_bioRE).

## Abstract

Automatic biomedical relation extraction (bioRE) is an essential task in biomedical research in order to generate high-quality labelled data that can be used for the development of innovative predictive methods. However, building such fully labelled, high quality bioRE data sets of adequate size for the training of state-of-the-art relation extraction models is hindered by an annotation bottleneck due to limitations on time and expertise of researchers and curators. We show here how Active Learning (AL) plays an important role in resolving this issue and positively improve bioRE tasks, effectively overcoming the labelling limits inherent to a data set. Six different AL strategies are benchmarked on seven bioRE data sets, using PubMedBERT as the base model, evaluating their area under the learning curve (AULC) as well as intermediate results measurements. The results demonstrate that uncertainty-based strategies, such as Least-Confident or Margin Sampling, are statistically performing better in terms of F1-score, accuracy and precision, than other types of AL strategies. However, in terms of recall, a diversity-based strategy, called Core-set, outperforms all strategies. AL strategies are shown to reduce the annotation need (in order to reach a performance at par with training on all data), from 6% to 38%, depending on the data set; with Margin Sampling and Least-Confident Sampling strategies moreover obtaining the best AULCs compared to the Random Sampling baseline. We show through the experiments the importance of using AL methods to reduce the amount of labelling needed to construct high-quality data sets leading to optimal performance of deep learning models. The code and data sets to reproduce all the results presented in the article are available at https://github.com/oligogenic/Deep_active_learning_bioRE.

## Introduction

With the expansion of the biomedical literature, many efforts have been made, either by improving search processes or automatic identification of relevant content [1], to aid researchers and clinicians in navigating this overwhelming amount of data and generating data

**Funding:** This work was supported by the Service Public de Wallonie Recherche by DIGITALWALLONIA4.AI [2010235—ARIAC to C.N. and T.L.]; the European Regional Development Fund (ERDF); an F.N.R.S-F.R.S CDR project [35276964 to T.L.]; Innoviris Joint R&D project Genome4Brussels [2020 RDIR 55b to T.L.]; and the Research Foundation-Flanders (F.W.O.) Infrastructure project associated with ELIXIR Belgium [I002819N to T.L.]. Computational resources have been provided by the Consortium des Equipements de Calcul Intensif (CECI), funded by the Fonds de la Recherche Scientifique de Belgique (F.R.S.-FNRS) under Grant No. 2.5020.11 and by the Walloon Region. The funders had no role in study design, data collection and analysis, decision to publish, or preparation of the manuscript.

**Competing interests:** The authors have declared that no competing interests exist.

resources that can be used for further study [2]. Research in biomedical text-mining and natural language processing has been shown to be key to supporting such biocuration, facilitating to a certain extent the population of biomedical knowledge resources.

Serving as a building block in such curation activities, biomedical Relation Extraction (bioRE) aims to detect and classify relations between biomedical entities within a text. It is often paired with the Named Entity Recognition, as it first needs to identify relevant entities [3]. While early relation extraction (RE) relied on template and rule-based methods [4], traditional machine learning methods involving features such as re-occurring terms or the structure of the sentences have emerged, offering a first foray into automated text-mining approaches for such a task [5–8]. New advances and state-of-the-art performances have been afterwards obtained with deep learning techniques [9]: convolutional and recurrent neural networks radically increased the accuracy for bioRE by exploiting the latent dependencies between words [10–14]. Additionally, graph neural networks (GNNs) that incorporated rich linguistic features, both within and between sentences [15–17] within the graph, provided an alternative to boost performance. Nonetheless, the current best models are adaptations of the *Bidirectional Encoder Representation for Transformer* (BERT) architecture [18], requiring only additional training or fine-tuning with biomedical papers to make them relevant for the biomedical domain [19–22]. An overview of pre-trained transformer-based biomedical language models is reported in Kalyan et al. [23], which also highlights the challenges of the use of deep learning models, of which one is the amount of data needed for retraining.

Luo et al. recently published an overview of existing bioRE data sets, showcasing the need for high-quality, complex data sets [24]. At the moment, most of the RE data sets are focused on single binary relations, i.e. relations involving two entities of specific types only, such as protein-protein interactions or chemical-disease interactions, found at the single sentence level. Challenges thus remained to train systems to find multiple RE types, RE involving more than two entities, as well as RE across multiple sentences. With BioRED, the authors partially filled that gap by creating a data set with diverse types of binary relations involving multiple entity types at the document level. While this type of data set promotes the development of more robust and accurate bioRE models, the authors have noted that one of its limitations is its size. Indeed, they could only include 600 abstracts as manual annotation is costly in time and expertise, a recurring problem for all RE data sets. However, due to the increasing use of deep learning methods for bioRE, large amounts of data are needed more often to achieve good performances.

As labelling additional samples in data sets may be difficult and substantial unlabelled data is readily available, alternative approaches are required that somehow sample optimally the unlabelled cohort data to train high-quality RE models without having to explicitly label all of the data.

Active learning (AL) is a key approach that provides a way to minimise the annotation cost [25]. It is achieved by training a model with a partially labelled data set and then the AL process itself will actively select among the unlabelled samples the set of samples it considers to be the most informative. This may be achieved for example by choosing the samples for which the model is the most uncertain about their predictions (prediction-based) or by selecting the data the most different from what it has already been trained on (diversity-based). An oracle, normally human, provides the correct labelling for these samples, which are subsequently added to the training set. The cycle begins anew with the newly obtained training set to train the model until a stopping criterion, such as a minimal performance of the trained model obtained on the test set or a number of newly labelled instances, is reached (Fig 1). Using such techniques minimises the amount of data to label to reach optimal performance.

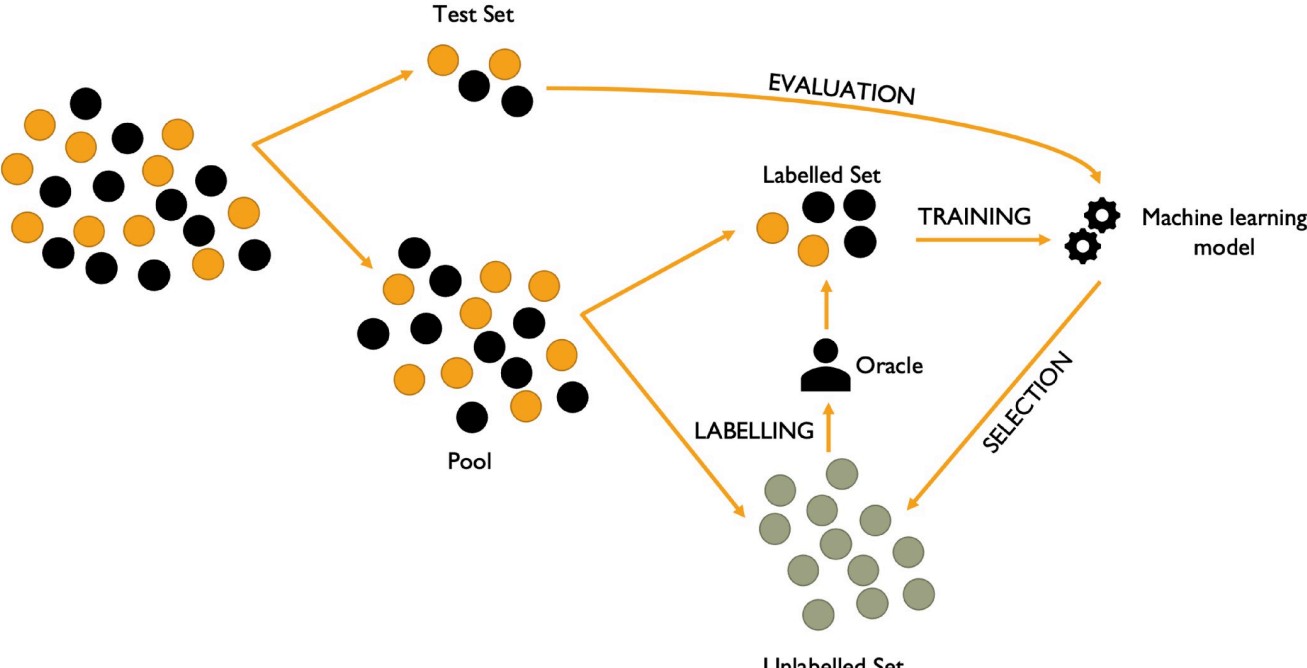

**Fig 1. Diagram representing an AL scenario.** The data set can be split into the test set, used to evaluate the performance of the machine learning model after each AL iteration, the labelled set and the unlabelled set. An active learning iteration is divided into 3 steps: training, selection and labelling. First, the labelled set is used to train a model, which is then used to select in the unlabelled set the most informative instances to label according to a strategy, such as the instances the model is the most uncertain about. Finally, those selected instances are labelled by an oracle (generally a human expert) and added to the labelled set. The AL loop can be stopped once a stop criterion is reached, such as a specific performance of the trained model on a test set, a number of AL iterations or a specific amount of instances that are to be labelled.

AL has already been extensively studied for several text-mining tasks, such as NER [26–32] or text classification [33–36], both for general and clinical domains, as well as for traditional [26, 28, 30, 32] or deep learning models [31, 33, 34, 36, 37], and it has been shown that the maximum accuracy may be reached with only a subset of the total data set. While AL has been studied for some bioRE data sets with simpler traditional machine learning models (e.g. random forests) [30, 38, 39], it has not yet been bench-marked with cutting-edge RE models, such as BERT-based models. Nonetheless, AL may prove to be an essential component in such data-hungry approaches.

In this work, we present a first study of the use of AL for labelling bioRE data sets, using a deep learning model (PubMedBERT) as a learner. We show that AL can reduce the need for labelling for bioRE and that strategies based on the predictions of the model allow the model to reach faster an optimal performance in terms of F1-score, accuracy and precision than by randomly labelling. We also demonstrate that a diversity-based strategy, called Core-set, out-performs all other AL strategies in terms of recall. We conclude this article by presenting some guidelines for the use of AL in real-world scenarios for labelling bioRE data sets.

## Materials and methods

### Description of the data sets

In this work, we evaluate seven different bioRE data sets, covering several types of entities (i.e., gene, protein, chemicals, drug and variant) and containing relations at the sentence and

**Table 1. Statistics about the RE data sets used in the experiments.**

| Name | Type of relation | # Instances | | | # Tokens per instance | |
|---|---|---|---|---|---|---|
| | | Total | Positive | Negative | Mean | Max |
| AIMED [42] | Protein-Protein | 5834 | 1000 | 4834 | 50.92 | 244 |
| BioRED* [24] | All binary relations between Gene, Chemical, Disease and Variant | 31586 | 5960 | 25626 | 392.5 | 761 |
| CDR* [43] | Chemical-Disease | 30056 | 7249 | 22807 | 299.44 | 748 |
| ChemProt [44] | Chemical-Gene | 45048 | 9950 | 35098 | 57.05 | 374 |
| DDI [45] | Drug-Drug | 33508 | 4999 | 28509 | 74.49 | 240 |
| Nary—DGV [46] | Drug-Gene-Variant | 6987 | 3407 | 3580 | 100.55 | 750 |
| Nary—DV [46] | Drug-Variant | 6087 | 3131 | 2956 | 82.24 | 1933 |

* indicates the data sets for which we created the negative set of instances

abstract levels. Each instance consists of the raw text, the entities, and the relation between those entities. This work focuses on binary relations, with the positive class corresponding to the presence of a relation and the negative class to the absence of the relation.

As a pre-processing step, we perform entity masking, where each entity is masked with a type in the text, e.g. *@GENE$* for an entity belonging to the gene type, as was done previously in similar studies using PubMedBERT [20, 24, 40]. This ensures that the classification of the relationship relies on the context around the entities and not the entities themselves, producing a better generalisation of the model [41]. When only positive instances are available for the data set, negative instances are generated by creating false relationships between entities with no existing connection in the data set. It should be noted that this process may produce an imbalanced data set biased towards the negative class, which has to be addressed in the bioRE task.

The general statistics of the investigated data sets are summarised in Table 1. Pre-processed data sets, and the scripts to import and convert data, can be found at https://github.com/oligogenic/Deep_active_learning_bioRE/tree/main/data.

## Active learning selection strategies

AL is conducted either in a stream-based or pool-based scenario. The stream-based scenario considers each unlabelled instance independently for labelling, while in the pool-based scenario, all the unlabelled instances are evaluated at the same time and only the top best queries, also called active batch, are selected for labelling [25]. In this work, the pool-based scenario is used as all the unlabelled instances are available at the start of the process and the data set will not change dynamically during the process, as is implied in a stream-based scenario.

As the data sets used in this work are fully annotated, the oracle is simulated, avoiding in this way any additional manual labelling [47].

The instances to be labelled are sampled according to a variety of strategies, which are classified in several categories according to how they characterise the instances as most interesting to be included in the training set: diversity-based, model-based and prediction-based. A noteworthy method not included in the above categories is the Random Sampling method. This method is often used as a baseline as it mimics the real-life settings where each instance has an equal probability of being selected while labelling and samples are not selected in a biased way.

Six strategies were selected for the current analysis, including Random Sampling as a baseline, covering the diversity-based and prediction-based categories, as well as one hybrid method combining the two categories. No model-based methods were chosen for this study as

they require excessive computational resources [48]. These methods were selected for their common uses as baselines in similar study of AL benchmarks in text classification and sentiment analysis [33, 36].

**Diversity-based instance sampling.**   Diversity-based methods, also called density-based or data-based methods, rely only on properties of the raw data to select the instances, e.g. by choosing the most diverse set of instances.

The *Core-set method* [49] was selected, as it is considered a standard diversity-based method in deep AL. In this method, the selected batch of unlabelled instances is generated to be the most different from the already labelled training instances as well as the most diverse among themselves. First, the distance between all the instances is computed as the Euclidean distance between the embeddings obtained from the last layer of the transformer. Then, an unlabelled instance, the furthest from the labelled instances, is selected to be added to the labelled set. This operation is repeated until the desired number of unlabelled instances is added. The aim of this approach is to minimise the distance between the labelled set and the unlabelled set.

**Prediction-based instance sampling.**   Prediction-based methods use the outputs of the model to score the instances. The instances with the highest scores, e.g. deemed as the most uncertain according to an uncertainty measure, are selected for labelling [25].

Three prediction-based methods using different uncertainty measures were chosen for the current study based on their frequency of use in the literature [50].

In the following equations, we denote the instance as $x$, the number of classes as $C$, $\hat{y}_i$ is the $i$-th most likely class predicted for the instance $x$ and $P(\hat{y}_i|x)$ is a probability-like predicted class distribution obtained for the instance $x$ for the class $\hat{y}_i$.

First, the *Least-Confident Sampling method* measures the difference between the most confident prediction and 100% confidence [51]. It is calculated as follows:

$$LC(x) = 1 - P(\hat{y}_1|x)$$

Second, the *Margin Sampling method* takes the difference between the top two most confident predictions [52, 53]. In our case, the method simply uses the difference between the predictions for the first ($\hat{y}_1$) and the second ($\hat{y}_2$) most likely class probabilities.

$$MS(x) = P(\hat{y}_1|x) - P(\hat{y}_2|x)$$

Finally, the *Entropy Sampling method* computes the difference between all predictions, as defined by information theory [54, 55].

$$ES(x) = -\sum_{i=1}^{C} P(y_i|x)\log(P(y_i|x))$$

**Hybrid instance sampling.**   As hybrid method, BatchBALD [56] was selected as it is known to perform particularly well for Deep Learning models [57]. BALD (Bayesian Active Learning by Disagreement) selects points which have the highest mutual information between model predictions and parameters $\omega$ of the model. This is estimated with the following equation:

$$\mathbb{I}(y_i; x, \omega) = \mathbb{H}(y_i|x) - \mathbb{E}_{p(\omega)}[\mathbb{H}(y_i|x, \omega)]$$

with $\mathbb{H}$ being the conditional entropy of the model's prediction for the class $y_i$, i.e. the general uncertainty of the model, and $\mathbb{E}$ representing the expected value of the entropy of the model prediction over the posterior of the model parameters $\omega$, i.e. the expected uncertainty for a given sampling of the model parameters. So the score will be high when the model is highly

uncertain for the prediction (high $\mathbb{H}$) while having many draws of the model parameters disagreeing which is the best way to explain the prediction (low $\mathbb{E}$), meaning a high score can be translated as a sample with high uncertainty and high disagreement between the model parameters to explain the prediction.

BatchBALD builds on the BALD acquisition function by computing the mutual information for a set of instances (i.e., a batch) instead of a unique instance. In that case, the expectation $\mathbb{E}$ can be approximated using a Monte Carlo estimator to sample parameter distribution of the model. In our experiments, 10 inference cycles are performed following the specifications in the original article [56].

## PubMedBERT model

PubMedBERT is a pre-trained transformer-based biomedical language model [20]. It follows the classical BERT architecture, based on a transformer, and was pre-trained using purely biomedical texts from PubMed, both abstracts and full-text articles. It is considered one of the state-of-the-art biomedical natural language processing models and has been deployed in a variety of studies in the last years [24, 58, 59].

For the RE task, we used a sentence classification framework with PubMedBERT. Different hyper-parameters are used for each of the data sets (see Table 2). Some of the parameters, controlling the training of the model, such as the maximum length of the input allowed, the number of training epochs and the learning rate hyper-parameters are tuned using the information in Gu et al. [20] and Lai et al. [60]. The rest of the hyper-parameters of the model, such as the optimiser, were kept as in Gu et al. [20]. This choice has been made as we want to study the AL process, which does not need an optimisation of the hyper-parameters. Hyperparameter optimisation can be done once the final labelled data set has been obtained and one desires the optimal performance for that data set.

The other hyper-parameters, i.e. the size of the training seed and the active batch size, controlling the active learning process, have been chosen so that after ten iterations of the AL process the whole pool is used (Table 2).

## Experimental settings

A $k$-fold cross-validation was used ($k = 5$), (Algorithm 1—Line 1) and the selection strategies were repeated $n = 3$ times (Algorithm 1—Line 3) with different training seeds, giving a total of 15 executions for each pair of strategy and data set. Overall, 6,300 fine-tuning experiments of the PubMedBERT model were performed (7 data sets × 5 folds × 3 seeds × (1 base model + (6 strategies × 10 iterations))). The experiments were conducted on a server with Ubuntu Desktop 20.04.5 LTS (GNU/Linux 5.15.0–56-generic x86_64) operating system, Nvidia driver 470.161.03, CUDA version 11.4, with 32GB RAM on 2 Asus GTX 1080 TI GPUs.

**Table 2. Hyper-parameters used for the different data sets.**

| Name | Max length | Number of epochs | Learning rate | Size training seed | Active batch size |
|---|---|---|---|---|---|
| AIMED | 256 | 10 | 1e-5 | 500 | 500 |
| BioRED | 512 | 3 | 2e-5 | 2500 | 2500 |
| CDR | 512 | 3 | 2e-5 | 2500 | 2500 |
| ChemProt | 256 | 3 | 2e-5 | 3600 | 3600 |
| DDI | 256 | 3 | 2e-5 | 2500 | 2500 |
| Nary | 512 | 10 | 1e-5 | 500 | 500 |

The main scripts for the benchmark used the libraries from huggingface: transformers, evaluate and accelerate [61]; as well as the DISTIL (https://github.com/decile-team/distil) library for the active learning methods. Statistical tests were conducted in R with the package scmamp and in Python with scipy.

A pseudocode for the process can be found in Algorithm 1. Performance measurements were computed after each training step of the model, by evaluating the trained model on an independent test set. It should be noted that the model is fine-tuned from the start at each iteration. The average of the results at each iteration is used for downstream analysis.

**Algorithm 1** Active learning analysis framework

```
Require: k ≥ 1 and n ≥ 1 and seed_size ≥ 1
 1: for (pool,test) in k-fold stratified data set splits do
 2:   repeat ← 1
 3:   for i = 0 to n do
 4:     labelled ← RandomSampling(seed_size, pool)
 5:     unlabelled ← pool\labelled
 6:     for all strategy in strategies do
 7:       AL_analysis(labelled, unlabelled, test, strategy)
 8:     end for
 9:   end for
10: end for
```

For the performances measures, given the binary classification setting (i.e., in RE, presence/absence of a relationship), the F1-score, precision, recall and accuracy metrics were used, as described below:

$$accuracy = \frac{(TP + TN)}{(TP + TN + FP + FN)} \tag{1}$$

$$precision = \frac{TP}{(TP + FP)} \tag{2}$$

$$recall = \frac{TP}{(TP + FN)} \tag{3}$$

$$\text{F1-score} = \frac{2 * (\text{precision} * \text{recall})}{(\text{precision} + \text{recall})} \tag{4}$$

where $TP$ is True Positives, $TN$ is True Negatives, $FP$ is False Positives and $FN$ represents the False Negatives. When dealing with imbalanced data sets, it is important to assess multiple performance metrics such as precision, recall, and F1-score. Accuracy can be misleading because it does not account for the disproportionate class distribution, leading to inflated scores for the majority class. Precision and recall are more informative, especially for the minority class, as they measure the model's ability to correctly identify positive instances. The F1-score provides a balanced measure of precision and recall, and is particularly useful for overall model evaluation on imbalanced data sets. Therefore, it is important to evaluate models using multiple metrics to obtain a comprehensive understanding of their performance on imbalanced data sets.

## Evaluating the active learning performance

Evaluating the active learning performances is done by (1) analysing visually the learning curves of the active learning processes, where it is expected that the learning curves of AL strategies will be above the learning curve of the Random baseline, signifying that they outperform it, by (2) observing the relative difference in their performance compared to Random Sampling

across the iterations, by (3) establishing the size of the subset of the data set needed to be labelled to reach a specific performance, and by (4) conducting statistical comparisons between the AL strategies. AL processes can also be observed through the distribution of classes in the labelled set across the iterations.

The recommendations of Reyes et al. [62] for conducting statistical comparisons of AL strategies were followed: AL selection strategies are compared either according to the analysis of the area under the learning curve (i.e. the curve obtained by measuring the metrics at each AL iteration), or by analysing the intermediate results (AL iterations) with non-parametric ranking statistical tests. This form of analysis has also been used previously in Ein-Dor et al. [36] and He et al. [63].

**Relative difference with the Random baseline.** The difference between the performance of the AL strategy and the Random baseline is computed at each AL iteration, matching the fold and repeat of the performance measures. The results are reported as a box plot per data set demonstrating the distribution of the relative differences with the Random baseline for each strategy for each iteration of the AL process.

**Analysis of the area under the learning curve.** The area under the learning curve (AULC) computes the area below the performance curve. The higher the AULC, the better the performance of the model.

With $m$ as the number of AL iterations, $z_i$ as the performance of the model at the $i$-th iteration and $L_i$ as the labelled set at the $i$-th iteration, the AULC of a learning strategy $\theta$ can be approximated with the trapezoidal rule [62], as

$$AULC_\theta = \frac{1}{2} \sum_{i=1}^{m-1} (z_{i+1} + z_i)(|L_{i+1}| - |L_i|) \tag{5}$$

The AULCs between two sampling strategies are statistically compared by computing the AULCs of each strategy for each data set and then ranking each strategy for each data set. The lower the rank, the higher the performance. We can test if a strategy performs significantly better with consistency across the data sets using a Friedman test [64], a non-parametric statistical test analysing the variance of ranks, as advised in Garcia et al. and Reyes et al. [62, 65]. The null hypothesis assumes that all strategies have identical performances.

The AULCs can be compared either directly with the Random baseline as control, in that case, we apply the Hommel post-hoc procedure [66] to correct the obtained $p$-values, or pairwise with all the possible comparisons, in that case, the Bergmann-Hommel post-hoc procedure is applied [67].

**Intermediate results analysis.** With the analysis of the AULC, important information derived from the intermediate results, i.e. the AL iterations, can be lost as it aggregates all the iterations into one unique value.

To obtain a value representing the difference between two strategies at a specific iteration, the cut-point scoring scheme is used [62], defined here as:

$$c_{i,j}^{\alpha,\beta} = z_{i,j}^\alpha - z_{i,j}^\beta \tag{6}$$

with $\alpha$ and $\beta$ being two selection strategies under comparison, and $z_{i,j}^\theta$ the performance realised by the selection strategy $\theta$ at the $i$-th iteration for the $j$-th data set, resulting in a matrix of cut-point scores.

If the dominant strategy $\alpha$ outperforms the strategy $\beta$, the difference in performance should be positive and increase during the AL process. Thus, this can be tested with a Page Trend test [68] where the ideal ranking for the cut-point scores is $(m, m-1, \cdots, 1)$. The null hypothesis corresponds to the case where the ranking is randomly ordered.

A statistical test is conducted for each pair of strategies with their matrix of cut-point scores. Each cell $c_{i,j}$ of the resulting table contains the p-value of the statistical test between the strategies associated with row $i$ and column $j$. A low p-value means that the probability that the ranking is random is also low and thus that the row strategy outperforms the column strategy across the data sets.

It should be noted that for this analysis, the first and last iterations, corresponding to the performances for the training seed and the performance with a full data set, were discarded in order to keep only the intermediate iterations where differences between sampling strategies should be observed. As the size of the active batch is around one-tenth of the total size of the pool set, some data sets had an extra iteration to reach the total size of the pool. Consequently, we discarded that extra iteration in order to have the same number of observations for all the data sets.

**Performance for a subset of data set.**   The average of the performance reached for the full data set was computed across all folds and repeats. It is then used as a threshold to determine for each sampling strategy the percentage of the data set needed to at least reach that performance.

**Class imbalance analysis.**   Class imbalance is an important problem in most biomedical data sets, with few positive examples. Studying the impact of the selection strategy relative to class imbalance is important in order to choose the best AL strategy according to the class distribution in the data set [33, 69, 70]. To better understand how selection strategies work with class imbalance, class imbalance is measured at each iteration of the AL process with a modified version of the Shannon Entropy, as done in Angeli et al. [33]:

$$Balance = \frac{-\sum_{i=1}^{C} \frac{c_i}{n} * \log\left(\frac{c_i}{n}\right)}{\log C} \tag{7}$$

where $C$ is the total number of classes, $c_i$ is the number of instances belonging to the class $i$ and $n$ is the total number of instances. A balance of 0 represents a completely imbalanced data set (i.e., a data set with instances belonging to a unique class) and 1 a perfectly balanced data set (i.e., a data set where each class possesses the same number of instances).

We also study the balance of the data sets by computing the fraction of positive examples in the training set. Resulting class imbalance values are averaged over each iteration per pair of data set and strategy.

## Results

### AL sampling strategies outperform Random Sampling

In Fig 2, we show the analysis of the relative difference of their F1-score with the Random baseline (see Methods Section Relative difference with the Random baseline). Except for Batch-BALD, all the AL strategies outperform the Random baseline. The difference in performance decreases the closer the training set is to the full data set, which is because the difference between the subset selected by an AL strategy and the Random baseline is disappearing.

While these results show that AL strategies perform better, the data sets of AIMED and CDR (Fig 2) are apparently more difficult to use with AL strategies: As can be observed the average AL strategy result is often close to random and the AL strategy variation shows that both good and bad outcomes, relative to the Random baseline, are obtained. Exploring the reasons for this problem provided no conclusive answers.

Visual inspection of the performance curves of AL processes reveals an inconsistency in the performances for those datasets across the iterations, i.e. random drops in the performance, more particularly in the case of recall and F1-score (S2 File).

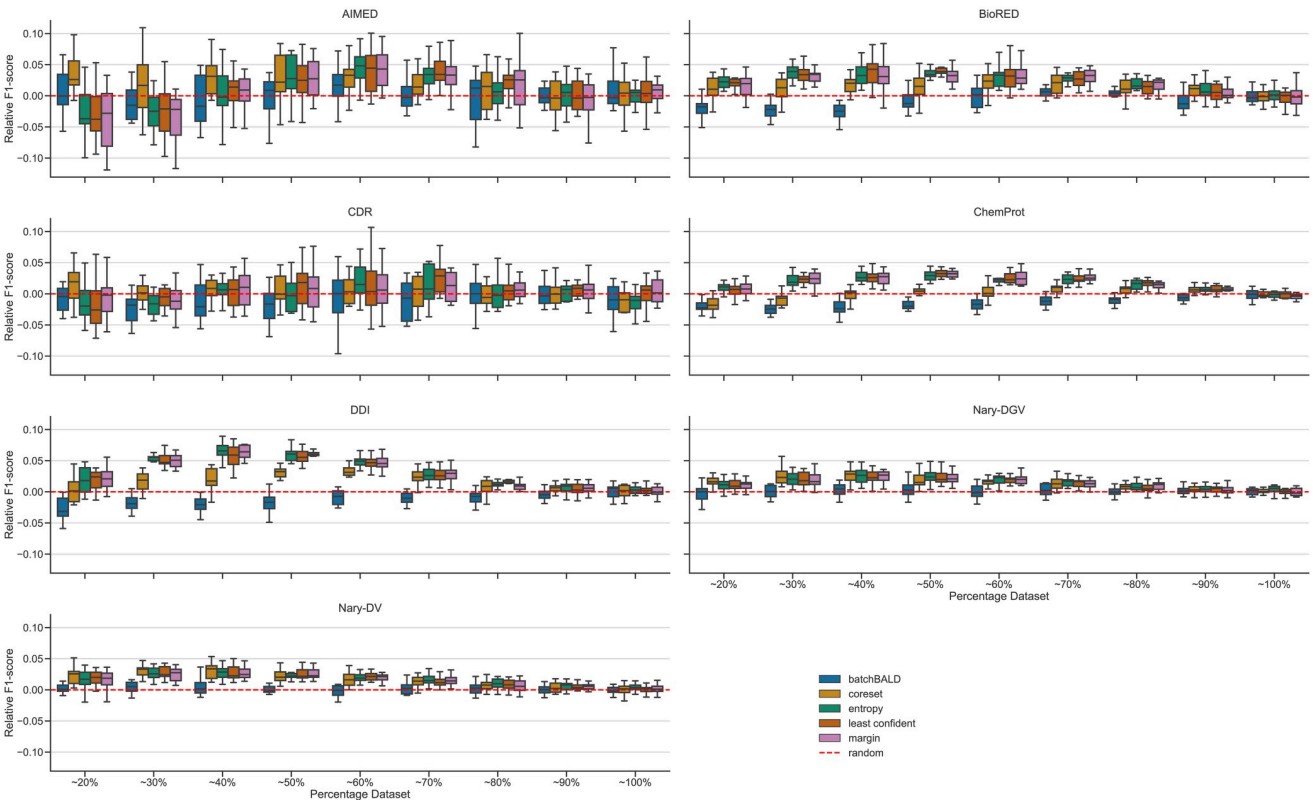

**Fig 2. Distribution of the relative difference between the AL strategies and the Random baseline across the AL iterations.** Y-values bigger than 0 indicate that the selection technique is performing better than the Random baseline. Except for BatchBALD, all AL strategies tend to have a positive difference compared to random This difference decreases with the increase of the size of the data set used for training. Results outside of 1.5*inter- quartile range from the first quartile and third quartile are removed for clarity. Boxplots containing the outliers are available in the S1 File.

## Uncertainty-based methods are preferred for AL

To evaluate further the performance of the different AL strategies, the AULC for each pair of a strategy and a data set was computed (see Methods Section Analysis of the area under the learning curve, Table 3).

After the visual inspection of the relative difference with the Random baseline (Fig 2), uncertainty-based methods, i.e. Entropy, Least-Confident and Margin Sampling strategies, are expected to perform better. It is observed that they are generally the methods with the highest AULC, with the exception on the AIMED data set, where the Core-set strategy performs better. The Random baseline has a better AULC than BatchBALD for the AIMED, BioRED, CDR and DDI data sets, a better AULC than Entropy Sampling for the BioRED and CDR data sets, and a better AULC than Core-set for the BioRED data set. The Least-Confident and Margin Sampling strategies have an overall better AULC than Random Sampling. The average rank for each strategy is computed by averaging the ranking obtained by the strategy across each data set, the lowest being the best ranking (see Methods Section Analysis of the area under the learning curve). The average rank of the strategies as shown in Table 3 reveals that all AL strategies, except for BatchBALD, have a better performance than the Random baseline. Interestingly, Random Sampling has a generally lower standard deviation than the AL strategies, which may be due to the fact that it uniformly chooses samples.

**Table 3. Average AULC (as in Eq 5) and standard deviation over the folds for each pair of data set and selection strategy based on F1-score.**

| Data set | BB | CS | E | LC | M | R |
|---|---|---|---|---|---|---|
| AIMED | 1590.4 ± 122.1 | **1699.3 ± 121.6** | 1627.9 ± 133.1 | 1641.4 ± 149.3 | 1639.3 ± 145.9 | 1612.7 ± 103.2 |
| BioRED | 10 328.8 ± 321.8 | 10 415.4 ± 423.3 | 10 131.0 ± 450.2 | 10 547.2 ± 533.0 | **10 591.9 ± 341.5** | 10 461.4 ± 259.6 |
| CDR | 10 317.5 ± 499.2 | 10 468.3 ± 748.5 | 10 110.4 ± 707.4 | 10 557.1 ± 726.8 | **10 596.9 ± 656.45** | 10 466.1 ± 324.9 |
| ChemProt | 21 301.0 ± 231.3 | 21 392.4 ± 906.8 | 21 724.1 ± 755.3 | **21 914.5 ± 798.3** | 21 567.8 ± 691.0 | 21 251.4 ± 390.1 |
| DDI | 15 530.9 ± 160.0 | 16 233.8 ± 185.8 | **16 624.6 ± 196.3** | 16 572.0 ± 201.1 | 16 594.4 ± 201.6 | 15 825.3 ± 231.1 |
| Nary—DGV | 3683.6 ± 18.3 | 3752.1 ± 22.3 | **3755.3 ± 7.8** | 3750.5 ± 14.1 | 3751.5 ± 14.2 | 3677.3 ± 9.2 |
| Nary—DV | 3552.0 ± 24.2 | 3617.5 ± 27.2 | **3619.3 ± 25.8** | 3617.1 ± 27.9 | 3615.3 ± 24.0 | 3543.8 ± 28.9 |
| Average rank | 5.286 | 2.857 | 3.000 | 2.429 | 2.429 | 5.000 |
| P-value | 0.775 | 0.068 | 0.091 | 0.041 | 0.041 | - |

Best AULC per data set are in bold. Average rank corresponds to the average over the column of the ranking of the strategies for each dataset. The lower the rank, the better. P-value is the result of the Friedman statistical test with a Hommel post-hoc procedure compared with the Random Sampling baseline. BB = BatchBALD, CS = Core-set, E = Entropy, LC = Least-Confident, M = Margin, R = Random.

When comparing the AL strategies with Random Sampling as a control, only Least-Confident and Margin Sampling strategies show a significant statistical difference ($p < 0.05$). However, when performing pairwise comparisons between the AL strategies, all the strategies except Entropy Sampling (S1 Fig) outperform BatchBALD with significant statistical differences.

In Table 4, the size of the labelled data set needed to reach the average performance that can be reached with a fully labelled data set was computed for each sampling strategy. A decrease of annotation between 6% to 38% is noted, with AIMED and CDR having the worst reduction. The Least-Confident and Margin Sampling strategies obtained an optimal performance with only a subset of the full data set, confirming the previous observations for the uncertainty-based methods.

## Different strategies optimise different metrics

To obtain a more precise insight into the performance of the different AL strategies, the intermediate results and the corresponding performance metrics are analysed at each iteration of the AL process (see Methods Section Intermediate results analysis).

**Table 4. Percentage of data set needed to reach at least the average performance with the total data set for each AL strategy.**

| Data set | BB | CS | E | LC | M |
|---|---|---|---|---|---|
| AIMED | 100 | 100 | 100 | 100 | 100 |
| BioRED | 99 | 89 | **69** | **69** | **69** |
| CDR | **94** | **94** | 100 | **94** | **94** |
| ChemProt | 80 | 100 | **70** | **70** | **70** |
| DDI | 100 | 84 | 75 | 75 | **65** |
| Nary—DGV | 89 | 81 | **72** | **72** | 81 |
| Nary—DV | 92 | **62** | **62** | **62** | **62** |

Best percentage of annotated data set per data set are in bold. BB = BatchBALD, CS = Core-set, E = Entropy, LC = Least-Confident, M = Margin

**Table 5. P-values for the Page trend statistical test with intermediate results based on F1-score.**

|  | BB | CS | E | LC | M | R |
|---|---|---|---|---|---|---|
| BB | - | 1.000 | 1.000 | 1.000 | 1.000 | 0.931 |
| CS | < **0.001** | - | 0.181 | 0.175 | 0.221 | < **0.001** |
| E | < **0.001** | 0.825 | - | 0.552 | 0.221 | < **0.001** |
| LC | < **0.001** | 0.831 | 0.458 | - | 0.229 | **0.001** |
| M | < **0.001** | 0.786 | 0.786 | 0.779 | - | **0.001** |
| R | 0.072 | 0.000 | 0.000 | 0.999 | 0.999 | - |

Dominant strategies are in the row labels, i.e. the hypothesis tested is the strategy in the row performs better than the strategy in the column. P-values below 0.05 are in bold. BB = BatchBALD, CS = Core-set, E = Entropy, LC = Least-Confident, M = Margin, R = Random.

Table 5 lists the *p*-values for the Page-Trend statistical tests with the F1-score as the performance measure between each pair of AL strategies. Page-Trend tests if the ranking of the cut-point scores of two strategies is positively increasing with the AL iterations, while the null hypothesis assumes a random order (see Methods Section Intermediate results analysis).

With the exception of BatchBALD, all the AL strategies statistically perform better than Random Sampling. Moreover, all AL strategies also outperform BatchBALD. No statistically significant difference is distinguished between the other AL strategies.

Interestingly, the analysis of the intermediate results with the accuracy as the performance measure (S1 Table) confirms the results obtained using the F1-score as performance metric, with the addition of Entropy and Least-Confident Sampling also outperforming the Core-set strategy.

The intermediate results based on the precision in Table 6 display a clear superiority of the uncertainty-based methods over the other types of methods. This underlines how accurate the model is when it is predicting the positive class. Margin Sampling, the method which had generally the best AULCs based on its F1-score, is once again statistically dominated by both the Entropy and the Least-Confident Sampling methods. While the Core-set strategy is still better than the BatchBALD strategy, it does not statistically outperform the Random Sampling in terms of precision.

Finally, the intermediate results based on the recall in Table 7 display a remarkable dominance of the Core-set method over all the strategies. This shows that a great portion of the positive instances are predicted as positive. BatchBALD is outperformed by all the other AL strategies, except for Random Sampling. Entropy Sampling is the only other method next to the Core-set strategy that also performs better than Random Sampling in terms of recall.

**Table 6. P-values for the Page trend statistical test with intermediate results based on precision.**

|  | BB | CS | E | LC | M | R |
|---|---|---|---|---|---|---|
| BB | - | 0.992 | 1.000 | 1.000 | 1.000 | 1.000 |
| CS | **0.008** | - | 1.000 | 1.000 | 0.998 | 0.411 |
| E | < **0.001** | < **0.001** | - | 0.290 | **0.044** | < **0.001** |
| LC | < **0.001** | < **0.001** | 0.718 | - | **0.007** | < **0.001** |
| M | < **0.001** | **0.002** | 0.958 | 0.994 | - | < **0.001** |
| R | < **0.001** | 0.598 | 1.000 | 1.000 | 1.000 | - |

Dominant strategies are in the row labels, i.e. the hypothesis tested is the strategy in the row performs better than the strategy in the column. P-values below 0.05 are in bold. BB = BatchBALD, CS = Core-set, E = Entropy, LC = Least-Confident, M = Margin, R = Random.

**Table 7. P-values for the Page trend statistical test with intermediate results based on recall.**

|  | BB | CS | E | LC | M | R |
|---|---|---|---|---|---|---|
| BB | - | 1.0000 | 0.996 | 0.978 | 0.989 | 0.349 |
| CS | **< 0.001** | - | **0.002** | **0.001** | **0.001** | **< 0.001** |
| E | **0.004** | 0.998 | - | 0.331 | 0.561 | **0.022** |
| LC | **0.023** | 0.999 | 0.677 | - | 0.570 | 0.090 |
| M | **0.012** | 0.999 | 0.448 | 0.439 | - | 0.063 |
| R | 0.660 | 1.000 | 0.980 | 0.914 | 0.940 | - |

Dominant strategies are in the row labels, i.e. the hypothesis tested is the strategy in the row performs better than the strategy in the column. P-values below 0.05 are in bold. BB = BatchBALD, CS = Core-set, E = Entropy, LC = Least-Confident, M = Margin, R = Random.

In summary, all AL strategies except BatchBALD perform significantly better than Random Sampling based on the F1-score, as observed through the relative difference with Random (Fig 2) and the AULCs (Table 3). Core-set outperforms all the strategies when looking at the recall, whereas uncertainty-based strategies outperform the other AL strategies regarding in relation to precision. This means that one could choose among the AL strategies according to the metric that one prefers to optimise.

## AL strategies increase the fraction of positive instances in the training set

As both balanced and imbalanced data sets are used (see Table 1), an investigation into how the different AL strategies across the iterations handle the class distribution was performed (see Fig 3). We observed that Random Sampling and BatchBALD keep the initial balance of the data set throughout the AL process, whereas the other AL strategies improve the balance of the data sets, especially in the earlier AL iterations.

Concerning the balance measure results for the unbalanced data sets (Fig 3A and S3 File), all AL strategies, except for BatchBALD, apparently over-sample the minority class at the

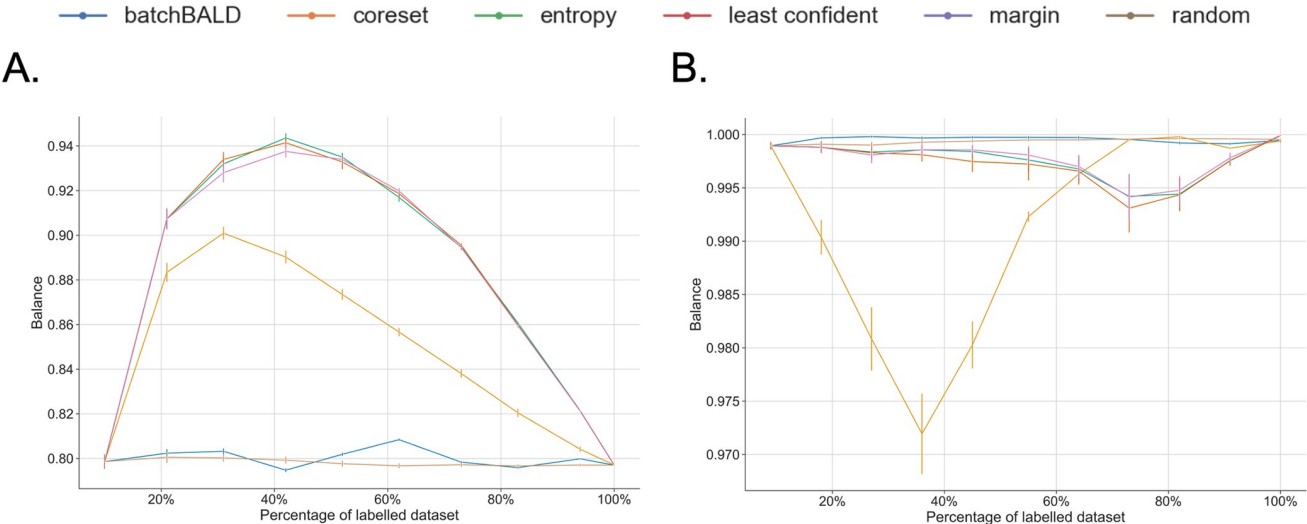

**Fig 3. Examples of balance measures using Shannon Entropy (see Eq 7) across AL iterations.** Results for (A) an unbalanced data set, CDR, and (B) a balanced data set, Nary-DGV. The measure provides an average over each iteration. The same behaviours were observed for the other data sets of the same distribution (S3 File). For each data set, each panel highlights the results for the specific AL strategy and shows the others in grey.

beginning of the learning process. This is reflected by the increase in the fraction of positive instances in the training set in the first half of the AL iterations. The uncertainty-based methods are notably more effective in increasing this balance than the Core-set method. Nonetheless, the Core-set selection method also seeks balance in the training set by increasing the fraction of positive instances, except for the ChemProt data set where it actually tends to select more negative instances and decreases the balance (Fig 4).

When dealing with balanced data sets (Fig 3B and S3 File), uncertainty-based and diversity-based strategies tend to slightly bias the training set. In the case of the Core-set strategy, it reduces the fraction of the positive instances in the first half of the AL process, whereas for the uncertainty-based strategies they increase the fraction of positive instances in the latter half of the process (Fig 5).

## Run-time analysis

We measured the time of execution for each AL strategy for the first two iterations for each data set, performing 5-fold cross-validation experiments, and reported the average time per sample selected from the unlabelled set for each iteration (see Table 8).

Uncertainty-based strategies (Entropy, Least-Confident and Margin) are the most time-efficient, as they only require to infer the unlabelled set of samples, whose size decreases throughout the AL process. BatchBALD runs inference 10 times on the unlabelled set, then is followed by a greedy approximation algorithm to create the batch. It means that it should also take less time to select samples across AL iterations, but it can initially take a lot of time to obtain the selected samples. Contrary to the previous strategies, Core-set needs to produce an embedding of all the samples at each iteration, regardless of the size of the labelled and unlabelled set. This operation is followed by a greedy k-center selection, so the computation time should not vary much across the AL iterations.

## Discussion

In this paper, it has been shown that AL strategies generally perform better than choosing randomly the instances to label in an iterative fashion in bioRE.

The visual inspection of the F1-score curves of the AL processes revealed some inconsistencies in the performances of the AL processes. This result aligns with earlier deep active learning classification experiments with BERT-based models [36, 71].

Nonetheless, the relative difference of the F1-score between the AL strategies and the Random baseline (Fig 2) clearly shows that AL strategies, except for BatchBALD, outperform the Random Sampling. The uncertainty-based strategies appear to be very successful in this respect. The difference between the AL strategies and the Random baseline obviously decreases with the ratio of the data set used for training, as the difference between the subsets selected by an AL strategy and the Random one decreases the closer the training set is to the full data set.

Additionally, the uncertainty-based methods achieve an optimal performance faster than other selection strategies, an observation shared with experiments on text classification with deep AL methods [34, 36, 50] and confirmed with the analysis of the AULCs, especially for the Least-Confident and Margin Sampling strategies (Table 3).

In an AL setting, one can decide to stop either when the trained model reaches a desired performance or when a fixed number of samples have been labelled, yet this threshold may depend on the data set itself: As one can observe in Fig 2 and Table 4, it is actually difficult to find a common threshold for all the data sets. The aim of AL in that case is to obtain an optimal performance with only a subset of the unlabelled data set. Moreover annotating unlabelled

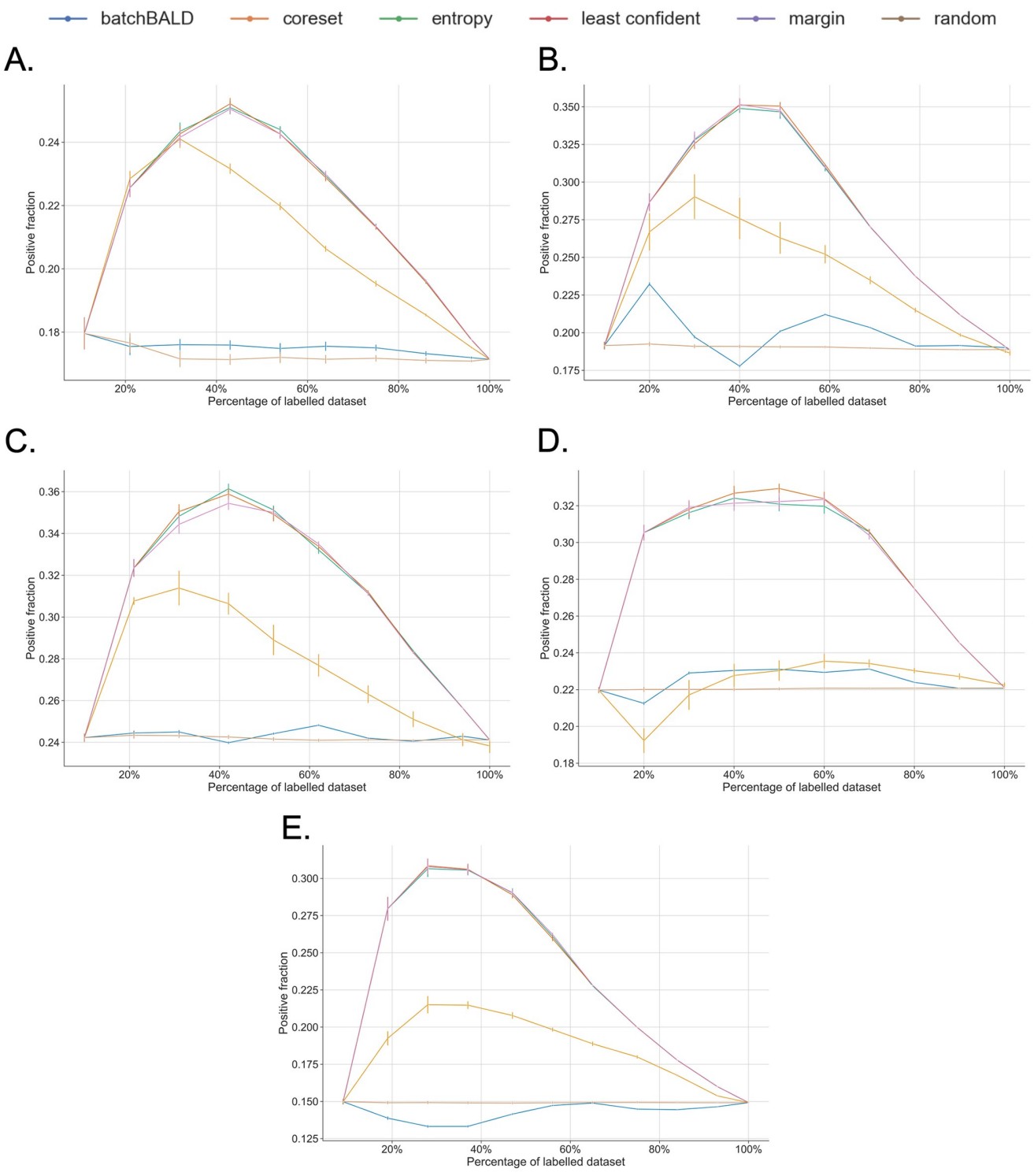

**Fig 4. Fraction of positive instances in the training set for the unbalanced data sets.** Data sets are as follows, (A) AIMED, (B) BioRED, (C) CDR, (D) ChemProt and (E) DDI. The measures are averaged over each iteration. For each data set, each panel highlights the results for a strategy and greys out the others.

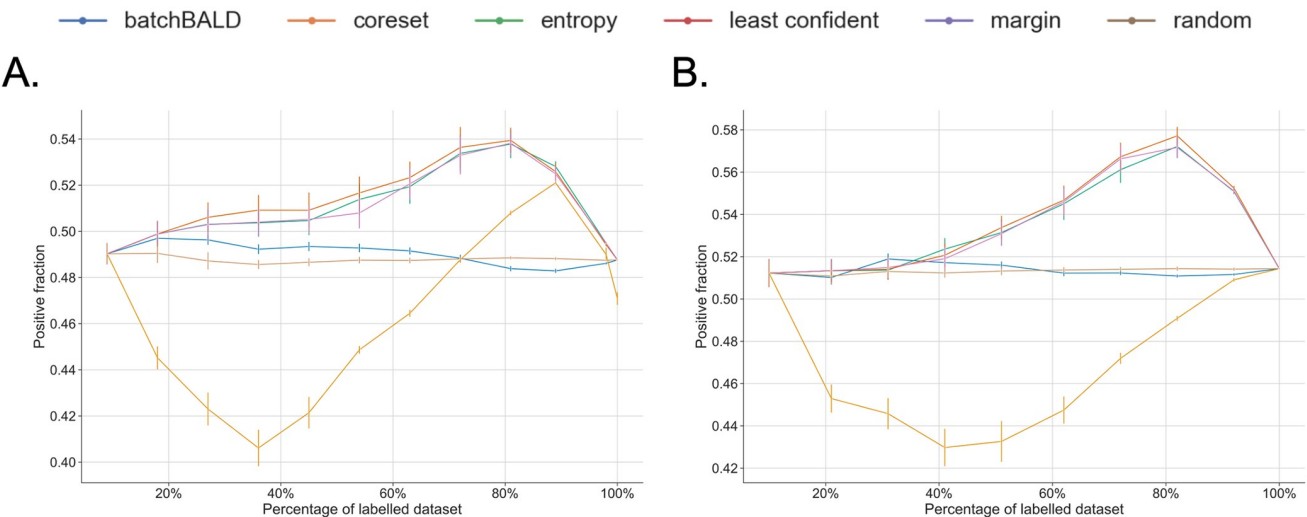

**Fig 5. Fraction of positive instances in the training set for the balanced data sets.** Data sets are as follows, (A) Nary-DGV and (B) Nary-DV. The measures are averaged over each iteration. For each data set, each panel highlights the results for a strategy and greys out the others.

instances may be costly, as for instance in bioRE, requiring limits on the amount of labelling to be imposed. One could also detect that a sufficient amount of labelling has been done by stopping when the performance improvement is lower than some minimal threshold.

The study of the intermediate results (Tables 5–7) refine our comparison between the strategies. Uncertainty-based methods optimise the accuracy and precision metrics, while Core-set, the diversity-based method, aims for the optimisation of the recall metric. According to our results, there is room for improvement towards selection methods that optimise precision and recall simultaneously. Generally, one would choose precision as the most relevant metric when one wants to be sure that what is predicted as positive is actually positive and recall when one wants to be sure to capture as many positive instances as possible. In bioRE, one could wish for example for better precision in the case of protein-protein interaction to ensure that one has access to accurate information, whereas one would choose recall for drug-drug interaction as this type of information could be essential for treatment and to avoid adverse drug reaction, so increasing the number of false positives would not be an hindrance in such case.

**Table 8. Execution time average over all the data sets per unlabelled sample for the two first iteration of the AL process for each AL strategy with their standard deviation.**

|  | 1st iteration | 2nd iteration |
|---|---|---|
| **BB** | 158.6 ± 39.1 ms | 153.7 ± 37.0 ms |
| **CS** | 13.7 ± 3.4 ms | 15.6 ± 2.9 ms |
| **E** | 10.4 ± 2.8 ms | 10.3 ± 2.8 ms |
| **LC** | 10.4 ± 2.8 ms | 10.3 ± 2.8 ms |
| **M** | 10.4 ± 2.8 ms | 10.3 ± 2.8 ms |
| **R** | $6.7 \times 10^{-5} \pm 2.5 \times 10^{-5}$ ms | $2.8 \times 10^{-5} \pm 2.1 \times 10^{-7}$ ms |

BB = BatchBALD, CS = Core-set, E = Entropy, LC = Least-Confident, M = Margin, R = Random. Results were rounded at the first decimal.

While one could think that a combination of both uncertainty and diversity strategies could lead to better performance based on these observations, BatchBALD, a hybrid method of both these types of strategies, performed overall even worse than Random Sampling. BatchBALD was however observed in their initial paper to not perform well for unbalanced data sets and to decrease in performance with the active batch size, which explains the bad performances in our experiments [56]. Another possibility is to explore methods that actually combine an uncertainty-based method with an approach such as Core-set, as was proposed in the filtered active submodular selection (FASS) method [72], or to create a new method exclusively for bioRE. In the latter case, the results of the current study may serve as a baseline for benchmarking this new method.

Additionally, the analysis of how the different AL strategies select the instances according to the class imbalance uncovered that uncertainty-based strategies had the tendency to sample more positive instances, especially in the earlier iterations of the AL process, improving the balance of the training set as a side effect. This leads to a better generalisation of the model and allows it to reach an earlier optimal performance in comparison to the other strategies. The effect of balancing the data set during the training phase of the AL iteration was not investigated, however it could have a positive impact on the performance of the model, at the cost of the size of the training data as we would need to undersample the majority class.

Finally, the run-time analysis of the different AL methods highlights the cost of using data-based and hybrid methods, which we show here are more computationally expensive, versus uncertainty-based methods, which are more time efficient. Our results argue in favour of the use of uncertainty-based AL strategies, especially as one should be concerned with both the economical and environmental costs of employing deep learning [73].

Our work focused on the empirical study for bioRE in a binary classification setting. A natural extension for this work would be the study of multi-class RE, as the identification of the relation type between entities is an important matter in bioRE [74]. While a wide variety of deep learning models were studied for bioRE, we explored in this work only BERT-based language models. An additional inquiry would be to explore other deep learning models, such as GNNs or generative large language models, which have proven to perform well on the RE task [75], to confirm our results for other deep learning architectures. Another direction for future works is the use of a self-supervision prior combined with AL, which has been shown to increase the performance of the AL process, as it uses the unlabelled data to obtain the underlying representation of the data set [76], or combined with the AL process in a semi-supervised setting [77]. Moreover, an extensive study on the mitigation of the class imbalance during the training or by using AL strategy aiming to balance the dataset [69, 70] are an additional path to explore as class imbalance is a major concern in bioRE. More investigation on these topics are necessary before being able to use AL for bioRE in the most efficient way.

## Conclusion

Building high-quality bioRE data sets requires both time and expertise in order to develop accurate RE tools to help with the biocuration of the increasing number of biomedical articles. AL methods aim to reduce the need of labelling by selecting the most informative instances to label among the unlabelled data. Knowing which AL methods perform the best with a specific type of data set allows for the development of annotation tools for bioRE using AL to efficiently select instances to be labelled by annotators, such as Paladin [78], AlpacaTag [79] or Label Sleuth [80] for text classification.

To the best of our knowledge, this work provides a first systematic study of bioRE with deep AL methods. With our experiments, it was shown that uncertainty-based methods,

especially the Least-Confident strategy, are the methods with overall the best performance, as was also observed in the case of text classification. However, if one aims to optimize recall in particular, the use of the Core-set selection strategy is advised.

## Supporting information

**S1 File. Boxplots of the relative difference of the F1-score between the AL strategies and the Random baseline with outliers.**
(PDF)

**S2 File. Performances curves of the accuracy, F1-score, precision and recall for the AL processes.**
(PDF)

**S3 File. Balance measures based on Shannon Entropy across active learning iterations for the AIMED, BioRED, ChemProt, DDI and Nary-DV data sets.**
(PDF)

**S1 Table. Page trend statistical test with intermediate results based on accuracy.** Dominant strategies are in the row labels, i.e. the hypothesis tested is the strategy in the row performs better than the strategy in the column. P-values below 0.05 are in bold.
(DOCX)

**S1 Fig. Results of the Friedman test and Bergmann–Hommel post-hoc procedure for the pairwise comparisons of the areas under the learning curve based on the F1-scores of the different active learning strategies and data sets.**
(EPS)

## Acknowledgments

The authors thank the Foundation 101 Genomes (f101g.org) for fruitful collaboration, creative exchange and scientific support. We would like to thank Barbara Gravel for her valuable comments during the writing of this manuscript.

## Author Contributions

**Conceptualization:** Charlotte Nachtegael, Tom Lenaerts.

**Data curation:** Charlotte Nachtegael.

**Formal analysis:** Charlotte Nachtegael, Jacopo De Stefani.

**Funding acquisition:** Tom Lenaerts.

**Investigation:** Charlotte Nachtegael.

**Methodology:** Charlotte Nachtegael, Jacopo De Stefani.

**Project administration:** Tom Lenaerts.

**Software:** Charlotte Nachtegael.

**Supervision:** Tom Lenaerts.

**Visualization:** Charlotte Nachtegael, Jacopo De Stefani, Tom Lenaerts.

**Writing – original draft:** Charlotte Nachtegael.

**Writing – review & editing:** Charlotte Nachtegael, Jacopo De Stefani, Tom Lenaerts.

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
