## [Decision Letter · Decision Letter 0]

21 Aug 2023

PONE-D-23-21026A study of Deep Active Learning methods to reduce labelling efforts in biomedical Relation ExtractionPLOS ONE

Dear Dr. Nachtegael,

Thank you for submitting your manuscript to PLOS ONE. After careful consideration, we feel that it has merit but does not fully meet PLOS ONE’s publication criteria as it currently stands. Therefore, we invite you to submit a revised version of the manuscript that addresses the points raised during the review process.

We look forward to receiving your revised manuscript.

Kind regards,

Wei Ju, Ph.D.

Academic Editor

PLOS ONE

Journal Requirements:

“This work was supported by the Service Public de Wallonie Recherche by DIGITALWALLONIA4.AI [2010235—ARIAC to C.N. and T.L.]; the European Regional Development Fund (ERDF); an F.N.R.S-F.R.S CDR project [35276964 to T.L.]; Innoviris Joint R&D project Genome4Brussels [2020 RDIR 55b to T.L.]; and the Research Foundation-Flanders (F.W.O.) Infrastructure project associated with ELIXIR Belgium [I002819N to T.L.]. The funders had no role in study design, data collection and analysis, decision to publish, or preparation of the manuscript.”

“Computational resources have been provided by the Consortium des Equipements de ´ 482 Calcul Intensif (CECI), funded by the Fonds de la Recherche Scientifique de Belgique ´ 483 (F.R.S.-FNRS) under Grant No. 2.5020.11 and by the Walloon Region. The authors 484 thank the Foundation 101 Genomes (f101g.org) for fruitful collaboration, creative 485 exchange and scientific support. We would like to thank Barbara Gravel for her 486 valuable comments during the writing of this manuscript”

“This work was supported by the Service Public de Wallonie Recherche by DIGITALWALLONIA4.AI [2010235—ARIAC to C.N. and T.L.]; the European Regional Development Fund (ERDF); an F.N.R.S-F.R.S CDR project [35276964 to T.L.]; Innoviris Joint R&D project Genome4Brussels [2020 RDIR 55b to T.L.]; and the Research Foundation-Flanders (F.W.O.) Infrastructure project associated with ELIXIR Belgium [I002819N to T.L.]. The funders had no role in study design, data collection and analysis, decision to publish, or preparation of the manuscript.

4. Please upload a copy of Figure 1-5 to which you refer in your text. If the figure is no longer to be included as part of the submission please remove all reference to it within the text

Reviewers' comments:

Reviewer's Responses to Questions

**Comments to the Author**

1. Is the manuscript technically sound, and do the data support the conclusions?

Reviewer #1: Partly

Reviewer #2: Yes

2. Has the statistical analysis been performed appropriately and rigorously? 

Reviewer #1: Yes

Reviewer #2: Yes

3. Have the authors made all data underlying the findings in their manuscript fully available?

Reviewer #1: Yes

Reviewer #2: Yes

4. Is the manuscript presented in an intelligible fashion and written in standard English?

Reviewer #1: Yes

Reviewer #2: Yes

5. Review Comments to the Author

Reviewer #1: This paper studies the problem of automatic biomedical relation extraction and leverage active learning to overcome the labelling limits inherent to a data set. The results demonstrate that uncertainty-based strategies, such as Least-Confident or Margin Sampling, are statistically performing better. This paper is well organized and clearly written. The technical details are also easy to follow. However, I have the following concerns:

1. GNNs have wide application in this task. Given generating labels needs huge effort, will graph-based semi-supervised learning help in this problem by leveraging extensive unlabeled data [1,2]? Please try to make a discussion.

2. I suggest authors to discuss the relationships between Large language models with this study.

3. I also suggest author to have a discussion with the graph active learning work [3].

[1] GHNN: Graph Harmonic Neural Networks for semi-supervised graph-level classification, Neural Network 22

[2] Dualgraph: Improving semi-supervised graph classification via dual contrastive learning, ICDE 22

[3] Focus on Informative Graphs! Semi-Supervised Active Learning for Graph-Level Classification, 2023

Reviewer #2: The paper highlights the significance of Active Learning (AL) in addressing the data annotation bottleneck for biomedical relation extraction (bioRE). It introduces six AL strategies, which are evaluated on seven distinct bioRE datasets using PubMedBERT as the base model. The study compares these strategies using various performance metrics, including F1-score, accuracy, precision, and recall. The results demonstrate the efficacy of AL in reducing annotation requirements while improving the performance of bioRE models.

The strengths of this paper lie in the following aspects:

1. The authors explore six different AL strategies, covering both uncertainty-based and diversity-based approaches. This comprehensive evaluation provides valuable insights into the strengths and weaknesses of each strategy in the context of bioRE tasks.

2. The paper conducts rigorous experiments on seven bioRE datasets, providing a robust assessment of the proposed AL strategies. The authors compare the strategies against a Random Sampling baseline and quantify their performance using multiple metrics, offering a well-rounded analysis.

3. The results indicate that uncertainty-based strategies, such as Least-Confident or Margin Sampling, outperform other strategies. This finding underscores the potential of AL to enhance model performance while reducing labeling efforts.

The weaknesses of this paper are as follows:

1. The paper focuses on six specific AL strategies, but it may benefit from discussing potential variations or combinations of these strategies that could further improve bioRE performance.

2. The paper does not carry out a comparative analysis of the time complexity and run-time performance of the various Active Learning (AL) methods examined.

3. A noteworthy shortcoming of the paper lies in its exclusive focus on comparing existing AL methods without introducing a novel AL approach of its own.

6. PLOS authors have the option to publish the peer review history of their article (what does this mean?). If published, this will include your full peer review and any attached files.

Reviewer #1: No

Reviewer #2: No

---

## [Author Response · Author response to Decision Letter 0]

11 Sep 2023

We would like to thank the reviewers for their comments. We hope that they will find our answers to their comments satisfactory and assuage their concerns.

---Reviewer #1---

"This paper studies the problem of automatic biomedical relation extraction and leverage active learning to overcome the labelling limits inherent to a data set. The results demonstrate that uncertainty-based strategies, such as Least-Confident or Margin Sampling, are statistically performing better. This paper is well organized and clearly written. 

We would like to thank the reviewer for the positive comments regarding our work. You can find below the answers to the questions and the requested corrections. All requested additions/changes are highlighted in the main manuscript with the changes tracked.

The technical details are also easy to follow. However, I have the following concerns:

1. GNNs have wide application in this task. Given generating labels needs huge effort, will graph-based semi-supervised learning help in this problem by leveraging extensive unlabeled data [1,2]? Please try to make a discussion."

We thank the reviewer for this interesting comment. We summarise our answer under the final point 3 below.

"2. I suggest authors to discuss the relationships between Large language models with this study."

We thank the reviewer for this interesting comment. We summarise our answer under the final point 3 below.

"3. I also suggest author to have a discussion with the graph active learning work [3]."

We agree with the reviewer that GNNs have been widely studied for biomedical relation extraction and have obtained good results. Biomedical relation extraction has been extensively studied with different deep learning models and up until recently, large language models performed well over other types of model. However, similar good results have been recently obtained with GNNs combined with knowledge graphs, as well as prompt-oriented and generative large language models. Such models would need to be studied extensively for the biomedical relation extraction task with also their own AL strategies, such as proposed by the reviewer in [3].

We also agree that unlabelled data could be leveraged in the AL process. Reference [4] din our paper shows how to use unlabelled data to pre-train the language model, but following the reviewer comments, we also added to the manuscript reference [5], presenting the possibility of using the unlabelled data in a semi-supervised setting, coupled with the AL process.

Following your comments, we added in the manuscript a discussion concerning the exploration of alternative deep learning models, as well as the link between our work and the use of large language models, in the discussion section on page 22. However, we did not add the proposed citations, as they are not directly relevant for our work and not in the scope of this paper, as we focus on BERT-based models for language representation.

---Reviewer #2---

"The paper highlights the significance of Active Learning (AL) in addressing the data annotation bottleneck for biomedical relation extraction (bioRE). It introduces six AL strategies, which are evaluated on seven distinct bioRE datasets using PubMedBERT as the base model. The study compares these strategies using various performance metrics, including F1-score, accuracy, precision, and recall. The results demonstrate the efficacy of AL in reducing annotation requirements while improving the performance of bioRE models.

The strengths of this paper lie in the following aspects:

1. The authors explore six different AL strategies, covering both uncertainty-based and diversity-based approaches. This comprehensive evaluation provides valuable insights into the strengths and weaknesses of each strategy in the context of bioRE tasks.

2. The paper conducts rigorous experiments on seven bioRE datasets, providing a robust assessment of the proposed AL strategies. The authors compare the strategies against a Random Sampling baseline and quantify their performance using multiple metrics, offering a well-rounded analysis.

3. The results indicate that uncertainty-based strategies, such as Least-Confident or Margin Sampling, outperform other strategies. This finding underscores the potential of AL to enhance model performance while reducing labeling efforts."

We would like to thank you for the positive evaluation of our work. You can find below the answers to the issues that you have raised. The requested additions/corrections are highlighted in blue in the manuscript, and corrections are in red and stroke through.

"The weaknesses of this paper are as follows:

1. The paper focuses on six specific AL strategies, but it may benefit from discussing potential variations or combinations of these strategies that could further improve bioRE performance."

We agree with the comment of the reviewer. In our work, we explored BatchBALD, a type of hybrid method that combines uncertainty- and diversity-based approaches, which did not give optimal results. However, BatchBALD is more complex than just a combination of the more simple methods we studied in our work. In the work of [6], they showed that a combination of uncertainty measures and finding the best submodular selection of the samples performs well on text classification, which could be worthwhile to explore in future studies. We added a reference to this topic in the discussion.

"2. The paper does not carry out a comparative analysis of the time complexity and run-time performance of the various Active Learning (AL) methods examined."

We thank the reviewer for this suggestion. We added a subsection in the results with experiments where we quantified the run time execution for the first two iterations of each AL selection strategy for the 5-fold of each data set. These experiments and subsequent discussion highlighted that uncertainty-based methods are the most computationally efficient, while BatchBALD and Core-set are more computationally expensive and time-consuming. This observation argues in favour of using uncertainty-based methods, as they obtain already good results and allow for a faster AL process.

"3. A noteworthy shortcoming of the paper lies in its exclusive focus on comparing existing AL methods without introducing a novel AL approach of its own."

We understand the comment of the reviewer and are aware of this limitation of our work. We chose to focus this paper on a benchmark of existing AL methods, trying to cover as much as possible the different categories of AL strategies with the aim to unearth guidelines for real-case use of AL for the biomedical relation extraction task and propose directions for future AL methods. In relation to the latter, we concluded that there is room for improvement towards methods that give both high precision and recall, as we show that the current methods mostly favour one of these two metrics. The creation and analysis of such a method is a different research question and would be a paper in itself. The results of the current study would serve as a baseline for comparison. We have added a clarification on this in the discussion section of the paper.

[1] GHNN: Graph Harmonic Neural Networks for semi-supervised graph-level classification, Neural Network 22

[2] Dualgraph: Improving semi-supervised graph classification via dual contrastive learning, ICDE 22

[3] Focus on Informative Graphs! Semi-Supervised Active Learning for Graph-Level Classification, 2023

[4] Margatina, Katerina, Barrault, Lo ¨ıc, Aletras, Nikolaos. On the Importance of Effectively Adapting Pretrained Language Models for Active Learning. ACL. 2022;doi:10.18653/v1/2022.acl-short.93.

[5] Gao M, Zhang Z, Yu G, Arık S ¨O, Davis LS, Pfister T. Consistency-Based Semi-supervised Active Learning: Towards Minimizing Labeling Cost. In: Vedaldi A, Bischof H, Brox T, Frahm JM, editors. Computer Vision – ECCV 2020. Cham: Springer International Publishing; 2020. p. 510–526

[6] Kai Wei, Rishabh Iyer, and Jeff Bilmes. 2015. Submodularity in data subset selection and active learning. In Proceedings of the 32nd International Conference on International Conference on Machine Learning - Volume 37 (ICML'15). JMLR.org, 1954–1963.

---

## [Decision Letter · Decision Letter 1]

20 Sep 2023

A study of Deep Active Learning methods to reduce labelling efforts in biomedical Relation Extraction

PONE-D-23-21026R1

Dear Dr. Nachtegael,

We’re pleased to inform you that your manuscript has been judged scientifically suitable for publication and will be formally accepted for publication once it meets all outstanding technical requirements.

Kind regards,

Shady Elbassuoni, PhD

Academic Editor

PLOS ONE

Additional Editor Comments (optional):

Reviewers' comments:

Reviewer's Responses to Questions

**Comments to the Author**

1. If the authors have adequately addressed your comments raised in a previous round of review and you feel that this manuscript is now acceptable for publication, you may indicate that here to bypass the “Comments to the Author” section, enter your conflict of interest statement in the “Confidential to Editor” section, and submit your "Accept" recommendation.

Reviewer #1: All comments have been addressed

Reviewer #2: All comments have been addressed

2. Is the manuscript technically sound, and do the data support the conclusions?

Reviewer #1: Yes

Reviewer #2: Yes

3. Has the statistical analysis been performed appropriately and rigorously? 

Reviewer #1: Yes

Reviewer #2: Yes

4. Have the authors made all data underlying the findings in their manuscript fully available?

Reviewer #1: Yes

Reviewer #2: Yes

5. Is the manuscript presented in an intelligible fashion and written in standard English?

Reviewer #1: Yes

Reviewer #2: Yes

6. Review Comments to the Author

Reviewer #1: This paper studies how to use deep active learning methods to reduce labelling efforts in biomedical relation extraction.

Authors have addressed all my concerns. So I recommend "accept".

Reviewer #2: I would like to thank the authors for taking the time and effort to address the concerns raised in my previous review. I appreciate the improvements made in the revised manuscript, which have significantly strengthened the quality of your work.

After reviewing the changes, I believe the manuscript is now suitable for publication. However, I would like to note that Figure 1 appears to be missing; I can only see the caption for the figure, but the figure itself is not present. I recommend addressing these minor points to further enhance the clarity and rigor of the paper.

7. PLOS authors have the option to publish the peer review history of their article (what does this mean?). If published, this will include your full peer review and any attached files.

Reviewer #1: No

Reviewer #2: No

---

## [Editor Report · Acceptance letter]

25 Sep 2023

PONE-D-23-21026R1 

A study of Deep Active Learning methods to reduce labelling efforts in biomedical Relation Extraction 

Dear Dr. Nachtegael:

I'm pleased to inform you that your manuscript has been deemed suitable for publication in PLOS ONE. Congratulations! Your manuscript is now with our production department. 

Kind regards, 

on behalf of

Dr. Shady Elbassuoni 

Academic Editor

PLOS ONE